# Effect of different treatment modalities on ovarian cancer patients with liver metastases: A retrospective cohort study based on SEER

**Na Li[1‡], Shanxiu Jin[2‡], Jingran Wu[2], Hongjuan Ji[3], Cheng Du[4]\*, Bona Liu[4]\***

**1** Department of Gynecology and Obstetrics, First Affiliated Hospital, Jilin University, Jilin, P. R. China, **2** Department of Oncology, General Hospital of Northern Theater Command, Dalian Medical University, Shenyang, P. R. China, **3** Department of Oncology, General Hospital of Northern Theater Command, Jinzhou Medical University, Shenyang, P. R. China, **4** Department of Oncology, General Hospital of Northern Theater Command, Shenyang, P. R. China

‡ NL and SJ are contributed equally to the work and are co-first authors.
\* apbnaliu@outlook.com (BL); doctor_ducheng@outlook.com (CD)

## Abstract

### Background

To examine the trends in morbidity and mortality among ovarian cancer patients with liver metastases, and investigate the impact of different treatments on both overall survival (OS) and cancer-specific survival (CSS).

### Methods

2,925 ovarian cancer patients with liver metastases from Surveillance, Epidemiology, and End Results 2010–2019 were included. The primary endpoint was considered as OS and CSS. We conducted trend analysis of the incidence, OS and CSS rates of liver metastases in ovarian cancer. Univariate and multivariate COX proportional risk models were used to investigate the association between different treatment methods and OS, and univariate and multivariate competing risk models were employed to evaluate the impact of treatment methods on CSS.

### Results

At the end of follow-up, 689 patients remained alive. The OS and CSS rates were 76.44% and 72.99% for all patients, respectively. There was a significant decreasing trend in the incidence [average annual percent change (AAPC) = -2.3, 95% confidence interval (CI): -3.9, -0.7], all-cause mortality (AAPC = -12.8, 95% CI: -15.6, -9.9) and specific mortality (AAPC = -13.0, 95% CI: -16.1, -9.8) rate of liver metastases in ovarian cancer. After adjusting all confounding factor, only receiving surgery was associated with OS [hazard ratio (HR) = 0.39, 95%CI: 0.31–0.48]/CSS (HR = 0.37, 95%CI: 0.30–0.47). Chemotherapy was found to be protective factor for OS (HR = 0.33, 95%CI: 0.30–0.37)/CSS (HR = 0.44, 95%CI: 0.39–0.50) of ovarian cancer patients, while not receiving surgery remained a risk factor. Additionally, the result of subgroup analyses also showed that only receiving surgery and

**Data Availability Statement:** The datasets used and/or analyzed during the current study are available at the https://github.com/apbnaliu/Effect-

of-different-treatment-modalities-on-ovarian-cancer-patients-with-liver-metastases.

**Funding:** This study was supported by Jilin Provincial Science and Technology Department Fund (20190201089JC), Liaoning Provincial Science and Technology Talents and Natural Science Foundation (2021-MS-042) and Jilin Provincial Department of Education Fund (JJKH20190002KJ). The funders had no role in study design, data collection and analysis, decision to publish, or preparation of the manuscript.

**Competing interests:** The authors have declared that no competing interests exist.

chemotherapy still were significant protective factor of OS and CSS for patients without other distant metastases, with distant metastases to the bone, lung, brain or other organs, with bone metastasis, and with lung metastasis.

## Conclusion

Our research has elucidated a downward trend in morbidity and mortality rates among patients with liver metastases originating from ovarian cancer. Only receiving surgery and chemotherapy as therapies methods confer survival benefits to patients.

## Introduction

Ovarian cancer is a prevalent gynecological malignancy and a leading cause of mortality in females [1]. Ovarian cancer possesses the potential to spread through tissue, lymph system, and blood [2]. Approximately 70% of ovarian cancer patients present with distant metastases at the time of diagnosis, resulting in an overall 5-year survival rate of less than 30% [3]. Epidemiological investigation indicates that the liver is the most frequent site of distant metastasis in ovarian cancer, followed by distant lymph nodes, lung, bone and brain [4]. The median survival time among patients with liver metastases was only 30 months [5], indicating a poor prognosis for ovarian cancer patients with liver metastases.

Previous studies have indicated that patients with advanced ovarian cancer who undergo debulking surgery have a more favorable prognosis compared to those who do not receive this treatment [6, 7]. The amount of residual disease following debulking surgery is a significant prognostic indicator for patients [8]. One study based on Surveillance, Epidemiology, and End Results (SEER) database showed that radiotherapy may be associated with a poorer prognosis in patients with primary ovarian cancer compared to those who do not receive radiation therapy [9]. In the study of Teckie et al., radiotherapy was considered an efficacious treatment for brain metastases of epithelial ovarian cancer [10]. In addition, prolonged delay in the initiation of adjuvant chemotherapy was associated with a decrease in overall survival (OS) rates for patients with advanced ovarian cancer [11]. However, to the best of our knowledge, the prognostic implications of various therapies on ovarian cancer patients with liver metastases remain unknown. It is imperative to investigate the optimal treatment for ovarian cancer patients with liver metastasis in order to enhance patient outcomes and alleviate disease burden.

Herein, the aim of this study was to examine the trends in morbidity and mortality among ovarian cancer patients with liver metastases using data from the SEER database, and to investigate the impact of different treatments on both OS and cancer-specific survival (CSS) in this population.

## Methods

### Population selection

This retrospective cohort study used data from the SEER database. The SEER database is a free access database, which compiles comprehensive information on cancer patients in the United States, encompassing demographics, tumor characteristics and details regarding mortality and survival rates [12, 13]. In this study, SEER*Stat software (version 8.4.0) was utilized to identify patients diagnosed with primary ovarian cancer from the SEER database 2010–2019.

The study population was required to meet the following inclusion criteria: (1) diagnosis of primary ovarian cancer based on International Classification of Diseases for Oncology codes (ICD-O-3); (2) age at diagnosis of 18 years or older; and (3) presence of liver metastases at the time of diagnosis. The exclusion criteria included: (1) patients with two or more primary cancers; (2) patients with missing information on surgery and follow-up. Finally, a total of 2,925 participants were included in this study. The process of selecting participants was illustrated in Fig 1. The requirement of ethical approval for this was waived by the Institutional Review Board of General Hospital of Northern Theater Command, because the data was accessed from SEER (a publicly available database). The need for written informed consent was waived by the Institutional Review Board of General Hospital of Northern Theater Command due to retrospective nature of the study. All methods were performed in accordance with the relevant guidelines and regulations.

## Data collection

The primary endpoint of this study was considered as the OS and CSS among ovarian cancer patients with liver metastases. OS was defined as the time from diagnosis to death from any cause, while CSS was defined as the time from the date of diagnosis to death from liver metastasis of ovarian cancer [14]. The following variables were extracted from the SEER database: age, race/ethnicity, marital status, income, grade (I, II, III, IV or unknown), tumor size, local lymph node metastasis, histologic (carcinosarcoma, clear cell, endometrioid, malignant brenner carcinoma, mucinous, serous and other), combined bone metastasis, combined brain metastasis, combined lung metastasis, combined other sites metastasis, cancer antigen-125 (CA-125), tumor location (only one side, bilateral), residual tumor volume, surgery, radiotherapy, chemotherapy, treatment (receiving surgery and radiotherapy, only receiving surgery, only receiving radiotherapy, and none).

## Statistical analysis

Continuous variables were presented as mean ± standard deviation (Mean ± SD, normal distribution) or median and quartile range [M (Q1, Q3), non-normal distribution]. Student's t-test or Mann-Whitney U test was used for between-group comparisons. The categorical variable was reported as the number of cases and composition ratio n (%), and group differences were compared using the chi-square test. $P$-value of less than 0.05 was deemed statistically significant.

We conducted a trend analysis of the incidence, OS and CSS among ovarian cancer patients with liver metastases in the SEER database from 2010 to 2019. Univariate Cox proportional risk model was utilized to identify potential covariates associated with OS ($P < 0.05$), while the univariate competing risk model was employed to identify potential covariates associated with CSS ($P < 0.05$). Subsequently, OS served as the outcome variable and different treatment methods were considered as independent variables, two models were established to investigate the association between different treatment methods and OS in ovarian cancer patients with liver metastases. Model 1 was univariate Cox proportional risk model (unadjusted covariates). Model 2 was multivariate Cox proportional risk model (adjusted all potential covariates associated with OS). Similarly, to assess the impact of various treatment methods on CSS, we developed univariate and multivariate competing risk models with CSS as the outcome and different treatment methods as independent variables. Model 1 was univariate competing risk model (unadjusted covariates). Model 2 was multivariate competing risk model (adjusted all potential covariates associated with CSS). The OS and CSS were calculated using the Kaplan-Meier curve. Subgroup analyses were conducted based on patients with different distant

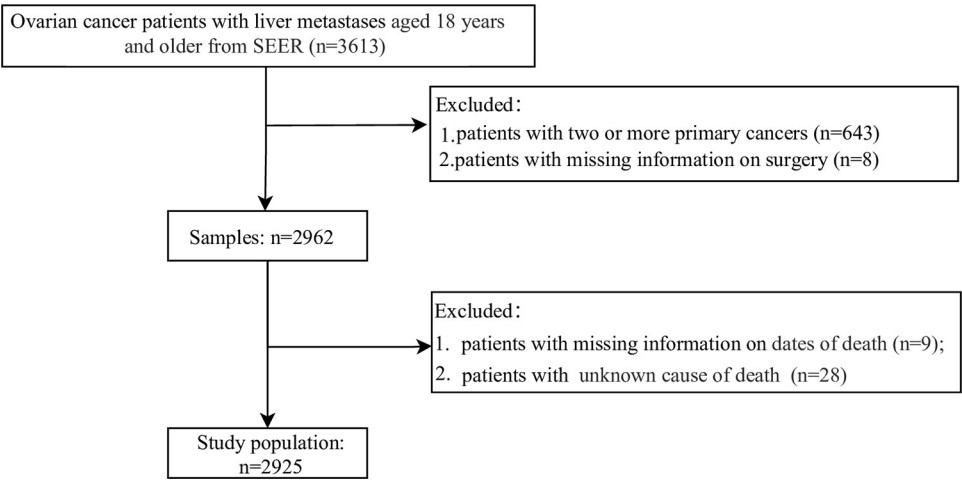

**Fig 1. The process of selecting participants.**

metastases types. In addition, we also developed online nomograms to predict patients' OS and CSS, respectively. The concordance index (C-index) was calculated to verify the predicting performance of two nomogram. The statistical analysis was conducted utilizing SAS 9.4 software (SAS Institute Inc., Cary, NC, USA). R software was utilized for the computation of the average annual percent change (AAPC) and its corresponding 95% confidence interval (CI).

## Results

### Patients' characteristics

A total of 2,925 ovarian cancer patients with liver metastases, with a mean age of 65.16 ± 13.49 years, were enrolled in this study. Most patients were white (78.46%). The median follow-up duration was 8.00 (1.00, 25.50) months, and at the conclusion of the follow-up period, a total of 689 patients remained alive. The OS and CSS rates were 76.44% and 72.99% for all patients, respectively. Detailed demographic and clinicopathological characteristics of all included patients were shown in Table 1.

### The incidence and survival trends of liver metastases in ovarian cancer

Fig 2 illustrates trend in the incidence and mortality of liver metastases in ovarian cancer. The overall age-adjusted incidence rate of liver metastases in ovarian cancer was decreasing trend with an AAPC value of -2.3 (95%CI: -3.9, -0.7). Simultaneously, we conducted an analysis of the trends in OS and CSS among ovarian cancer patients with liver metastases. All-cause mortality and liver metastasis of ovarian cancer-specific mortality trends were declined, with AAPC value of -12.8 (95% CI: -15.6, -9.9) and -13.0 (95% CI: -16.1, -9.8), respectively. In addition, S1 Fig. also shows the incidence trend of liver metastases in ovarian cancer stratified by age, tumor grade, and treatment modality. Although there was no statistical significance in certain subgroups, all incidence rate trend showed a tendency to a decrease in liver metastases in ovarian cancer stratified by age, tumor grade, and treatment modality. S2 Fig. also demonstrates a declining trend in the all-cause mortality of liver metastases in ovarian cancer stratified by age, tumor grade, and treatment modality. Similar results were observed for specific mortality of liver metastases in ovarian cancer in different subgroups (S3 Fig).

**Table 1. Demographic and clinicopathological characteristics of all included patients.**

| Variables | All patients (n = 2925) | Alive (n = 689) | Dead (n = 2236) | P |
|---|---|---|---|---|
| Age, year, Mean ± SD | 65.16 ± 13.49 | 60.84 ± 12.76 | 66.49 ± 13.43 | <0.001 |
| Race/ethnicity, n (%) | | | | <0.001 |
| Black | 346 (11.83) | 57 (8.27) | 289 (12.92) | |
| White | 2295 (78.46) | 532 (77.21) | 1763 (78.85) | |
| Other | 284 (9.71) | 100 (14.51) | 184 (8.23) | |
| Marital status, n (%) | | | | <0.001 |
| Married | 1254 (42.87) | 354 (51.38) | 900 (40.25) | |
| Not married | 1545 (52.82) | 301 (43.69) | 1244 (55.64) | |
| Unknown | 126 (4.31) | 34 (4.93) | 92 (4.11) | |
| Income, n (%) | | | | <0.001 |
| <$70,000 | 1859 (63.56) | 366 (53.12) | 1493 (66.77) | |
| ≥ $70,000 | 1066 (36.44) | 323 (46.88) | 743 (33.23) | |
| Grade, n (%) | | | | 0.391 |
| Grade I & Grade II | 97 (3.32) | 28 (4.06) | 69 (3.09) | |
| Grade III & Grade IV | 889 (30.39) | 202 (29.32) | 687 (30.72) | |
| Unknown | 1939 (66.29) | 459 (66.62) | 1480 (66.19) | |
| Tumor size, n (%) | | | | <0.001 |
| ≤50 | 375 (12.82) | 114 (16.55) | 261 (11.67) | |
| 50–100 | 561 (19.18) | 166 (24.09) | 395 (17.67) | |
| 100–200 | 573 (19.59) | 154 (22.35) | 419 (18.74) | |
| >200 | 80 (2.74) | 17 (2.47) | 63 (2.82) | |
| Unknown | 1336 (45.68) | 238 (34.54) | 1098 (49.11) | |
| Local lymph node metastasis, n (%) | | | | <0.001 |
| No | 758 (25.91) | 121 (17.56) | 637 (28.49) | |
| Yes | 566 (19.35) | 74 (10.74) | 492 (22.00) | |
| Unknown | 1601 (54.74) | 494 (71.70) | 1107 (49.51) | |
| Histologic, n (%) | | | | <0.001 |
| Carcinosarcoma | 111 (3.79) | 22 (3.19) | 89 (3.98) | |
| Clear cell | 76 (2.60) | 20 (2.90) | 56 (2.50) | |
| Endometrioid | 50 (1.71) | 16 (2.32) | 34 (1.52) | |
| Malignant Brenner Carcinoma | 931 (31.83) | 99 (14.37) | 832 (37.21) | |
| Mucinous | 69 (2.36) | 5 (0.73) | 64 (2.86) | |
| Serous | 1325 (45.30) | 475 (68.94) | 850 (38.01) | |
| Other | 363 (12.41) | 52 (7.55) | 311 (13.91) | |
| Combined bone metastasis, n (%) | | | | <0.001 |
| No | 2632 (89.98) | 661 (95.94) | 1971 (88.15) | |
| Yes | 179 (6.12) | 16 (2.32) | 163 (7.29) | |
| Unknown | 114 (3.90) | 12 (1.74) | 102 (4.56) | |
| Combined brain metastasis, n (%) | | | | <0.001 |
| No | 2776 (94.91) | 675 (97.97) | 2101 (93.96) | |
| Yes | 29 (0.99) | 1 (0.15) | 28 (1.25) | |
| Unknown | 120 (4.10) | 13 (1.89) | 107 (4.79) | |
| Combined lung metastasis, n (%) | | | | <0.001 |
| No | 2081 (71.15) | 568 (82.44) | 1513 (67.67) | |
| Yes | 709 (24.24) | 105 (15.24) | 604 (27.01) | |
| Unknown | 135 (4.62) | 16 (2.32) | 119 (5.32) | |
| Combined other sites metastasis, n (%) | | | | <0.001 |

*(Continued)*

**Table 1.** (Continued)

| Variables | All patients (n = 2925) | Alive (n = 689) | Dead (n = 2236) | P |
|---|---|---|---|---|
| No | 2182 (74.60) | 511 (74.17) | 1671 (74.73) | |
| Yes | 513 (17.54) | 151 (21.92) | 362 (16.19) | |
| Unknown | 230 (7.86) | 27 (3.92) | 203 (9.08) | |
| CA-125, n (%) | | | | <0.001 |
| Negative/normal/within normal limits | 73 (2.50) | 21 (3.05) | 52 (2.33) | |
| Positive/elevated | 2246 (76.79) | 577 (83.74) | 1669 (74.64) | |
| Unknown | 606 (20.72) | 91 (13.21) | 515 (23.03) | |
| Tumor location, n (%) | | | | 0.149 |
| Only one side | 1049 (35.86) | 263 (38.17) | 786 (35.15) | |
| Bilateral | 1876 (64.14) | 426 (61.83) | 1450 (64.85) | |
| Residual tumor volume, n (%) | | | | <0.001 |
| No gross residual tumor nodules | 350 (11.97) | 190 (27.58) | 160 (7.16) | |
| No cytoreductive surgery | 1462 (49.98) | 149 (21.63) | 1313 (58.72) | |
| Optimal debulking | 269 (9.20) | 91 (13.21) | 178 (7.96) | |
| Residual tumor nodule(s) greater than 1 cm | 157 (5.37) | 58 (8.42) | 99 (4.43) | |
| Macroscopic residual tumor nodule(s), size not stated | 159 (5.44) | 36 (5.22) | 123 (5.50) | |
| Unknown | 528 (18.05) | 165 (23.95) | 363 (16.23) | |
| Surgery, n (%) | | | | <0.001 |
| Debulking | 885 (30.26) | 375 (54.43) | 510 (22.81) | |
| Oophorectomy | 147 (5.03) | 56 (8.13) | 91 (4.07) | |
| Oophorectomy with omentectomy | 221 (7.56) | 76 (11.03) | 145 (6.48) | |
| Pelvic exenteration | 55 (1.88) | 26 (3.77) | 29 (1.30) | |
| Natural orifice surgery | 17 (0.58) | 2 (0.29) | 15 (0.67) | |
| None | 1600 (54.70) | 154 (22.35) | 1446 (64.67) | |
| Radiotherapy, n (%) | | | | 0.018 |
| Yes | 69 (2.36) | 8 (1.16) | 61 (2.73) | |
| None | 2856 (97.64) | 681 (98.84) | 2175 (97.27) | |
| Chemotherapy, n (%) | | | | <0.001 |
| Yes | 1981 (67.73) | 631 (91.58) | 1350 (60.38) | |
| No/Unknown | 944 (32.27) | 58 (8.42) | 886 (39.62) | |
| Treatment, n (%) | | | | <0.001 |
| None | 1557 (53.23) | 149 (21.63) | 1408 (62.97) | |
| Surgery and radiotherapy | 30 (1.03) | 3 (0.44) | 27 (1.21) | |
| Only receiving surgery | 1299 (44.41) | 532 (77.21) | 767 (34.30) | |
| Only receiving radiotherapy | 39 (1.33) | 5 (0.73) | 34 (1.52) | |
| Time, month, M (Q₁, Q₃) | 8.00 (1.00, 25.00) | 23.00 (8.00, 52.00) | 5.00 (1.00, 19.00) | <0.001 |
| Status, n (%) | | | | <0.001 |
| Alive | 689 (23.56) | 689 (100.00) | 0 (0.00) | |
| Dead of liver metastasis of ovarian cancer | 2135 (72.99) | 0 (0.00) | 2135 (95.48) | |
| Dead of other cause | 101 (3.45) | 0 (0.00) | 101 (4.52) | |

Abbreviations: CA-125 = cancer antigen-125; SD = standard deviation; M (Q1, Q3) = median and quartile range.

## Impact of different treatments on survival of liver metastases in ovarian cancer

Univariate Cox proportional risk and competing risk analyses demonstrated that age, race/ethnicity, marital status, income, grade, tumor size, local lymph node metastasis, histologic,

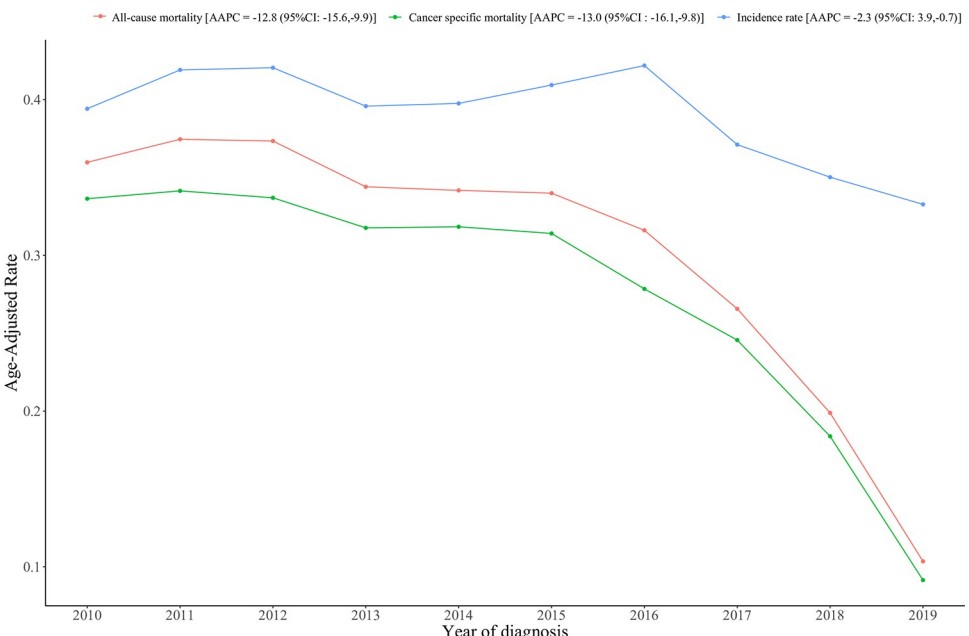

**Fig 2. The incidence rates and mortality of liver metastases in ovarian cancer between 2010 and 2019.**

combined bone metastasis, combined brain metastasis, combined lung metastasis, combined other sites metastasis, CA-125, residual tumor volume may be covariates associated with OS and CSS (*P*<0.05) (S1 Table). As shown in Table 2, after adjusting all covariates, ovarian cancer patients with liver metastases who only received surgery was associated with OS [Model 2: hazard ratio (HR) = 0.39, 95%CI: 0.31–0.48, *P*<0.001]. Using debulking as a reference, the absence of surgery was identified as a significant risk factor for OS (Model 1: HR = 4.23, 95% CI: 3.80–4.70, *P*<0.001; Model 2: HR = 2.66, 95%CI: 2.13–3.33, *P*<0.001), while the impact of other surgical methods on OS was not statistically significant (*P*>0.05). Also, chemotherapy was found to be a significant protective factor for OS of ovarian cancer patients with liver metastases, even after adjusting for variables (Model 2: HR = 0.33, 95%CI: 0.30–0.37, *P*<0.001). Similarly, after adjusting all covariates, in terms of CSS of ovarian cancer patients with liver metastases, only receiving surgery (Model 2: HR = 0.37, 95%CI: 0.30–0.47, *P*<0.001) and chemotherapy (Model 2: HR = 0.44, 95%CI: 0.39–0.50, *P*<0.001) were significant protective factor, while not receiving surgery (Model 2: HR = 2.68, 95%CI: 2.10–3.43, *P*<0.001) remained a risk factor. The Kaplan-Meier curve showed that only received surgery had a higher OS (S4A Fig) and CSS (S5A Fig). Surgical method also appeared to affect patient survival, with pelvic exenteration were found to have a higher OS (S4B Fig) and CSS than others (S5B Fig). Furthermore, patients who received radiotherapy may have lower OS (S4C Fig) and CSS (S5C Fig). The impact of chemotherapy on prognosis of patients with ovarian cancer was obvious, leading to significantly improved OS (S4D Fig) and CSS (S5D Fig) compared to those who do not receive this treatment.

## Subgroup analyses based on patients with different distant metastases types

In order to better explore the effects of treatment modalities on OS and CSS in ovarian cancer patients with liver metastases who had different characteristics, we conducted a stratified analysis based on different distant metastases types. As shown in Table 3, we found that only

**Table 2. Association between different treatment methods and OS/ CSS in ovarian cancer patients with liver metastases.**

| Outcomes | Variables | Model 1 | | Model 2 | |
|---|---|---|---|---|---|
| | | HR (95%CI) | *P* | HR (95%CI) | *P* |
| OS | Treatment | | | | |
| | None | Ref | | Ref | |
| | Surgery and radiotherapy | 0.72 (0.49–1.05) | 0.088 | 0.88 (0.57–1.35) | 0.545 |
| | Only receiving surgery | 0.24 (0.22–0.27) | <0.001 | 0.39 (0.31–0.48) | <0.001 |
| | Only receiving radiotherapy | 0.91 (0.65–1.28) | 0.597 | 0.72 (0.51–1.03) | 0.073 |
| | Surgery | | | | |
| | Debulking | Ref | | Ref | |
| | Oophorectomy | 1.19 (0.95–1.49) | 0.125 | 1.12 (0.89–1.41) | 0.347 |
| | Oophorectomy with omentectomy | 1.17 (0.97–1.40) | 0.100 | 1.15 (0.95–1.40) | 0.142 |
| | Pelvic exenteration | 0.78 (0.54–1.14) | 0.199 | 0.81 (0.55–1.18) | 0.267 |
| | Natural orifice surgery | 2.05 (1.23–3.43) | 0.006 | 1.65 (0.98–2.80) | 0.061 |
| | None | 4.23 (3.80–4.70) | <0.001 | 2.66 (2.13–3.33) | <0.001 |
| | Radiotherapy | | | | |
| | None | Ref | | Ref | |
| | Yes | 1.53 (1.19–1.97) | 0.001 | 1.07 (0.82–1.41) | 0.613 |
| | Chemotherapy | | | | |
| | No/Unknown | Ref | | Ref | |
| | Yes | 0.22 (0.21–0.25) | <0.001 | 0.33 (0.30–0.37) | <0.001 |
| CSS | Treatment | | | | |
| | None | Ref | | Ref | |
| | Surgery and radiotherapy | 0.76 (0.59–0.99) | 0.046 | 0.71 (0.49–1.04) | 0.077 |
| | Only receiving surgery | 0.30 (0.27–0.33) | <0.001 | 0.37 (0.30–0.47) | <0.001 |
| | Only receiving radiotherapy | 1.08 (0.80–1.45) | 0.628 | 0.85 (0.63–1.14) | 0.282 |
| | Surgery | | | | |
| | Debulking | Ref | | Ref | |
| | Oophorectomy | 1.20 (0.98–1.48) | 0.081 | 1.15 (0.94–1.40) | 0.180 |
| | Oophorectomy with omentectomy | 1.14 (0.97–1.35) | 0.115 | 1.13 (0.96–1.33) | 0.156 |
| | Pelvic exenteration | 0.81 (0.58–1.12) | 0.200 | 0.82 (0.60–1.14) | 0.247 |
| | Natural orifice surgery | 1.81 (1.16–2.81) | 0.009 | 1.61 (0.99–2.63) | 0.055 |
| | None | 3.42 (3.10–3.77) | <0.001 | 2.68 (2.10–3.43) | <0.001 |
| | Radiotherapy | | | | |
| | None | Ref | | Ref | |
| | Yes | 1.57 (1.31–1.90) | <0.001 | 1.11 (0.84–1.46) | 0.474 |
| | Chemotherapy | | | | |
| | No/Unknown | Ref | | Ref | |
| | Yes | 0.31 (0.28–0.34) | <0.001 | 0.44 (0.39–0.50) | <0.001 |

Abbreviations: OS = overall survival; CSS = cancer-specific survival; HR = hazard ratio; CI = confidence interval; Ref = reference. Model 1: crude model; Model 2: adjusted age, race/ethnicity, marital status, income, grade, tumor size, local lymph node metastasis, histologic, combined bone metastasis, combined brain metastasis, combined lung metastasis, combined other sites metastasis, cancer antigen-125, residual tumor volume

receiving surgery and chemotherapy still were significant protective factor of OS and CSS for patients without other distant metastases (Subgroup I), with distant metastases to the bone, lung, brain or other organs (Subgroup II), with bone metastasis (Subgroup III), and with lung metastasis (Subgroup IV). Additionally, for ovarian cancer patients with liver metastases who combined bone metastasis, surgery type, including oophorectomy with omentectomy

**Table 3. Subgroup analyses based on patients with different distant metastases types.**

| Population | Variables | OS | | CSS | |
|---|---|---|---|---|---|
| | | HR (95%CI) | *P* | HR (95%CI) | *P* |
| Subgroup I: Patients without other sites metastasis (n = 1556) | Treatment | | | | |
| | None | Ref | | Ref | |
| | Surgery and radiotherapy | 0.93 (0.42–2.07) | 0.860 | 0.98 (0.61–1.57) | 0.923 |
| | Only receiving surgery | 0.34 (0.25–0.47) | <0.001 | 0.39 (0.27–0.56) | <0.001 |
| | Only receiving radiotherapy | 0.38 (0.14–1.04) | 0.060 | 0.50 (0.15–1.65) | 0.256 |
| | Surgery | | | | |
| | Debulking | Ref | | Ref | |
| | None | 2.96 (2.17–4.05) | <0.001 | 2.67 (1.83–3.88) | <0.001 |
| | Oophorectomy | 1.03 (0.75–1.41) | 0.860 | 1.05 (0.80–1.39) | 0.725 |
| | Oophorectomy with omentectomy | 1.14 (0.89–1.46) | 0.311 | 1.15 (0.93–1.44) | 0.201 |
| | Pelvic exenteration | 0.77 (0.46–1.29) | 0.326 | 0.77 (0.49–1.20) | 0.244 |
| | Natural orifice surgery | 1.68 (0.85–3.34) | 0.136 | 1.54 (0.80–2.96) | 0.201 |
| | Radiotherapy | | | | |
| | None | Ref | | Ref | |
| | Yes | 0.90 (0.49–1.64) | 0.721 | 1.06 (0.46–2.43) | 0.897 |
| | Chemotherapy | | | | |
| | No/Unknown | Ref | | Ref | |
| | Yes | 0.32 (0.27–0.37) | <0.001 | 0.42 (0.35–0.50) | <0.001 |
| Subgroup II: Patients combined bone metastasis, brain metastasis, lung metastasis, other sites metastasis (n = 1092) | Treatment | | | | |
| | None | Ref | | Ref | |
| | Surgery and radiotherapy | 1.08 (0.60–1.93) | 0.804 | 0.69 (0.40–1.18) | 0.174 |
| | Only receiving surgery | 0.36 (0.25–0.53) | <0.001 | 0.29 (0.20–0.41) | <0.001 |
| | Only receiving radiotherapy | 1.10 (0.74–1.65) | 0.634 | 1.23 (0.90–1.68) | 0.187 |
| | Surgery | | | | |
| | Debulking | Ref | | Ref | |
| | None | 3.00 (1.98–4.55) | <0.001 | 3.43 (2.35–5.00) | <0.001 |
| | Oophorectomy | 1.13 (0.74–1.72) | 0.581 | 1.12 (0.77–1.63) | 0.563 |
| | Oophorectomy with omentectomy | 1.18 (0.84–1.67) | 0.337 | 1.09 (0.80–1.49) | 0.593 |
| | Pelvic exenteration | 0.77 (0.43–1.40) | 0.392 | 0.81 (0.46–1.46) | 0.489 |
| | Natural orifice surgery | 1.77 (0.63–4.96) | 0.275 | 2.03 (0.73–5.62) | 0.174 |
| | Radiotherapy | | | | |
| | None | Ref | | Ref | |
| | Yes | 1.55 (1.13–2.13) | 0.007 | 1.48 (1.11–1.97) | 0.007 |
| | Chemotherapy | | | | |
| | No/Unknown | Ref | | Ref | |
| | Yes | 0.29 (0.25–0.35) | <0.001 | 0.39 (0.32–0.47) | <0.001 |

(*Continued*)

**Table 3.** (Continued)

| Population | Variables | OS | | CSS | |
|---|---|---|---|---|---|
| | | HR (95%CI) | *P* | HR (95%CI) | *P* |
| Subgroup III: Patients combined bone metastasis (n = 179) | Treatment | | | | |
| | None | Ref | | Ref | |
| | Surgery and radiotherapy | 0.40 (0.13–1.22) | 0.108 | 0.34 (0.15–0.81) | 0.014 |
| | Only receiving surgery | 0.28 (0.11–0.72) | 0.008 | 0.28 (0.13–0.58) | <0.001 |
| | Only receiving radiotherapy | 0.86 (0.45–1.62) | 0.639 | 0.86 (0.51–1.45) | 0.566 |
| | Surgery | | | | |
| | Debulking | Ref | | Ref | |
| | None | 3.17 (1.04–9.65) | 0.042 | 2.21 (0.89–5.46) | 0.087 |
| | Oophorectomy | 2.79 (0.73–10.65) | 0.134 | 3.01 (0.80–11.31) | 0.102 |
| | Oophorectomy with omentectomy | 0.44 (0.15–1.29) | 0.135 | 0.42 (0.21–0.84) | 0.014 |
| | Pelvic exenteration | 0.51 (0.04–6.71) | 0.612 | 0.24 (0.06–0.98) | 0.047 |
| | Natural orifice surgery | 3.69 (0.37–37.28) | 0.268 | 3.15 (1.02–9.69) | 0.046 |
| | Radiotherapy | | | | |
| | None | Ref | | Ref | |
| | Yes | 0.92 (0.53–1.59) | 0.770 | 0.87 (0.54–1.39) | 0.555 |
| | Chemotherapy | | | | |
| | No/Unknown | Ref | | Ref | |
| | Yes | 0.43 (0.28–0.66) | <0.001 | 0.43 (0.29–0.66) | <0.001 |
| Subgroup IV: Patients combined lung metastasis (n = 709) | Treatment | | | | |
| | None | Ref | | Ref | |
| | Surgery and radiotherapy | 0.75 (0.36–1.56) | 0.441 | 0.47 (0.25–0.90) | 0.022 |
| | Only receiving surgery | 0.36 (0.22–0.58) | <0.001 | 0.32 (0.20–0.49) | <0.001 |
| | Only receiving radiotherapy | 0.85 (0.51–1.42) | 0.536 | 0.97 (0.70–1.33) | 0.838 |
| | Surgery | | | | |
| | Debulking | Ref | | Ref | |
| | None | 2.78 (1.65–4.70) | <0.001 | 2.91 (1.78–4.75) | <0.001 |
| | Oophorectomy | 1.06 (0.61–1.84) | 0.839 | 1.11 (0.71–1.75) | 0.639 |
| | Oophorectomy with omentectomy | 1.02 (0.66–1.57) | 0.928 | 0.94 (0.64–1.40) | 0.773 |
| | Pelvic exenteration | 1.10 (0.52–2.32) | 0.808 | 1.12 (0.67–1.87) | 0.674 |
| | Natural orifice surgery | 1.31 (0.39–4.36) | 0.657 | 1.42 (0.41–4.92) | 0.580 |
| | Radiotherapy | | | | |
| | None | Ref | | Ref | |
| | Yes | 1.18 (0.77–1.79) | 0.453 | 1.09 (0.77–1.53) | 0.637 |
| | Chemotherapy | | | | |
| | No/Unknown | Ref | | Ref | |
| | Yes | 0.27 (0.22–0.33) | <0.001 | 0.39 (0.32–0.49) | <0.001 |

Abbreviations: OS = overall survival; CSS = cancer-specific survival; HR = hazard ratio; CI = confidence interval; Ref = reference. Adjusted age, race/ethnicity, marital status, income, grade, tumor size, local lymph node metastasis, histologic, combined bone metastasis (not was adjusted in Subgroup II and III), combined brain metastasis (not was adjusted in Subgroup II), combined lung metastasis (not was adjusted in Subgroup II and IV), combined other sites metastasis (not was adjusted in Subgroup II), cancer antigen-125, residual tumor volume.

**Table 4. C-indexes for the nomograms.**

| Nomogram | C-index (95%CI) |
|---|---|
| OS predicting nomogram | 0.756 (0.746–0.766) |
| CSS predicting nomogram | 0.752 (0.742–0.763) |

(HR = 0.42, 95%CI: 0.21–0.84, $P$ = 0.014), pelvic exenteration (HR = 0.24, 95%CI: 0.06–0.98, $P$ = 0.047) and natural orifice surgery (HR = 3.15, 95%CI: 1.02–9.69, $P$ = 0.046) was considered to be associated with CSS.

## Establishment and validation of the nomogram

In univariate Cox proportional risk and competing risk analyses, age, race/ethnicity, marital status, income, grade, tumor size, local lymph node metastasis, histologic, combined bone metastasis, combined brain metastasis, combined lung metastasis, combined other sites metastasis, CA-125, and residual tumor volume were associated with both OS and CSS ($P<0.05$) (S1 Table). Thus, online OS (https://dynamic-nomogram-for-predicting-overall-survival—2023.shinyapps.io/DynNomapp/) and CSS (https://dynamic-nomogram-for-predicting-cancer-specific-survival—2023.shinyapps.io/DynNomapp/) predicting nomogram were established, respectively. C-index was used to verify the predicting performance of two nomogram (Table 4). Specifically, C-index of OS nomogram was 0.756 (95% CI: 0.746–0.766), and OS nomogram was 0.752 (95%CI: 0.742–0.763). These findings also indicated that two online nomograms had a good predicting value for OS and CSS in patients with ovarian cancer liver metastases.

## Discussion

It is crucial to explore diverse treatment options for patients with liver metastases from ovarian cancer in order to improve patient prognosis. In this study, utilizing data from the SEER 2010–2019 registry, we examined the trends in morbidity and mortality among patients with liver metastases from ovarian cancer. Additionally, only receiving surgery and chemotherapy were also found to be significant protective factor for OS and CSS of patients.

Liver metastases are a frequent occurrence in patients diagnosed with ovarian cancer and are widely recognized as the primary cause of mortality associated with this disease [15]. The findings of this study revealed a noteworthy decrease in the incidence of liver metastases among ovarian cancer patients in the United States between 2010 and 2019. Additionally, there was a significant decreasing trend in the all-cause mortality/ cancer-specific mortality between 2010 and 2019. This could be attributed to timely intervention, and advancements in therapeutic approaches [16]. Although there was no statistical significance in certain subgroups, all incidence rate, all-cause mortality, cancer specific mortality trend showed a tendency to a decrease in ovarian cancer with liver metastases when stratified by age, tumor grade, and treatment modality.

Distant metastatic sites have a significant impact on the OS of patients with ovarian cancer that has spread [5]. Treatment strategies may vary depending on the specific sites of metastasis. Numerous therapeutic approaches for ovarian cancer with different sites of metastasis are currently under extensive investigation. For example, a review has demonstrated that whole brain radiotherapy (WBRT) is a viable and efficacious treatment modality for ovarian cancer patients with brain metastases [17]. In the study conducted by Cao et al., it was reported that chemotherapy and surgery were associated with lung metastases from ovarian cancer [18], and there was no significant difference in radiotherapy between ovarian cancer patients with or without lung metastasis. Therapeutic approaches play a crucial role in guiding clinical practice [19, 20]; however, the prognostic impact of various therapeutic strategies on patients with ovarian cancer liver metastases remains unclear to date. In our research, we incorporated more clinicopathological information. After adjusting for all potential confounding factors, the findings revealed that only receiving surgery were associated with OS and CSS of patients. A study of 72 patients with brain metastases from ovarian cancer showed that the combination

of surgery and WBRT (median survival time: 23.07 months) resulted in superior survival outcomes compared to either surgery alone (median survival time: 6.90 months) or WBRT alone (median survival time: 5.33 months) [21]. Our study revealed no significant difference in the correlation between surgery combined with radiotherapy and OS/CSS among ovarian cancer patients with liver metastasis. Furthermore, in all subgroup analyses, only receiving surgery and chemotherapy also were beneficial for OS/CSS for patients. Similar to the findings in the general population, we observed no significant impact of different surgical method on patients' OS and CSS in subgroups without other distant metastases (Subgroup I), with distant metastases to the bone, lung, brain or other organs (Subgroup II), and with lung metastasis (Subgroup IV) ($P>0.05$). Notably, when comparing to Debulking as a reference, the absence of surgery posed a risk factor for both OS and CSS within these specific subgroups. This observation may be attributed to the limited sample size of our study. It is worth noting that oophorectomy with omentectomy, pelvic exenteration and natural orifice surgery were related to CSS for ovarian cancer patients with liver metastases who combined bone metastasis. These findings may also imply that the type of surgery has an impact on the prognosis of diverse patients. Further prospective studies with larger sample size should be conducted to confirm our findings.

Additionally, this study also proposed two online dynamic nomogram to predict OS and CSS among ovarian cancer patients with liver metastasis, respectively. Overall, two model exhibited a good prediction performance. In clinical practice, clinicians can utilize these user-friendly dynamic nomograms to make early intervention decisions, which may improve prognoses of patients.

There are several limitations to this study. Firstly, the SEER database solely documented the metastases of liver, brain, lung, bone and distant lymph nodes; hence the specific metastases of other sites such as peritoneal metastases remain unknown [22]. Secondly, variables pertaining to comorbidities, types of chemotherapy administered and adjuvant agents as well as the sequence of treatment were not extracted form SEER database [23]. Thirdly, due to the limited sample size, it was not possible to conduct subgroup analysis on patients with distant metastases to the liver and brain. Fourthly, as mentioned by Forte S et al., liver resection is feasible during either primary debulking surgery (PDS) or interval debulking surgery (IDS) [24]. But all the data were obtained from the SEER public database, which prevented us from distinguishing between the sequencing of patients' chemotherapy and surgery, as well as differentiating between PDS and IDS patients. Lastly, because the patient cohort in this study was limited to the US population, further validation of these findings is necessary across diverse populations worldwide.

## Conclusion

In summary, our research has elucidated a downward trend in morbidity and mortality rates among patients with liver metastases originating from ovarian cancer. Only receiving surgery and chemotherapy were protective factor for OS and CSS of patients. The findings of this study provide valuable guidance for clinicians and patients in selecting optimal treatment modalities to enhance the prognosis of individuals with liver metastases from ovarian cancer.

## Supporting information

**S1 Fig.** The incidence trend of liver metastases in ovarian cancer stratified by (a) age, (b) tumor grade, (c) surgery, (d) radiotherapy and (e) chemotherapy.
(PDF)

**S2 Fig.** The all-cause mortality trend of liver metastases in ovarian cancer stratified by (a) age, (b) tumor grade, (c) surgery, (d) radiotherapy and (e) chemotherapy.
(PDF)

**S3 Fig.** The specific mortality trend of liver metastases in ovarian cancer stratified by (a) age, (b) tumor grade, (c) surgery, (d) radiotherapy and (e) chemotherapy.
(PDF)

**S4 Fig. Kaplan-Meier curve of OS.**
(PDF)

**S5 Fig. Kaplan-Meier curve of CSS.**
(PDF)

**S1 Table. Screening of confounding factors.**
(DOCX)

## Author Contributions

**Conceptualization:** Na Li, Shanxiu Jin, Cheng Du, Bona Liu.

**Data curation:** Jingran Wu, Hongjuan Ji.

**Formal analysis:** Jingran Wu, Hongjuan Ji.

**Investigation:** Jingran Wu, Hongjuan Ji.

**Methodology:** Jingran Wu, Hongjuan Ji.

**Writing – original draft:** Na Li, Shanxiu Jin.

**Writing – review & editing:** Na Li, Shanxiu Jin, Cheng Du, Bona Liu.

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
