## [Decision Letter · Decision Letter 0]

16 Nov 2023

PONE-D-23-22973Effect of different treatment modalities on ovarian cancer patients with liver metastases: a retrospective cohort study based on SEERPLOS ONE

Dear Dr. Liu,

Thank you for submitting your manuscript to PLOS ONE. After careful consideration, we feel that it has merit but does not fully meet PLOS ONE’s publication criteria as it currently stands. Therefore, we invite you to submit a revised version of the manuscript that addresses the points raised during the review process.

We look forward to receiving your revised manuscript.

Kind regards,

Federico Ferrari, MD, PhD

Academic Editor

PLOS ONE

Journal Requirements:

"This study was supported by Jilin Provincial Science and Technology Department Fund (20190201089JC), Liaoning Provincial Science and Technology Talents and Natural Science Foundation (2021-MS-042) and Jilin Provincial Department of Education Fund (JJKH20190002KJ)."

"This study was supported by Jilin Provincial Science and Technology Department Fund (20190201089JC), Liaoning Provincial Science and Technology Talents and Natural Science Foundation (2021-MS-042) and Jilin Provincial Department of Education Fund (JJKH20190002KJ)."

"This study was supported by Jilin Provincial Science and Technology Department Fund (20190201089JC), Liaoning Provincial Science and Technology Talents and Natural Science Foundation (2021-MS-042) and Jilin Provincial Department of Education Fund (JJKH20190002KJ)."

Reviewers' comments:

Reviewer's Responses to Questions

**Comments to the Author**

1. Is the manuscript technically sound, and do the data support the conclusions?

Reviewer #1: Yes

Reviewer #2: Yes

2. Has the statistical analysis been performed appropriately and rigorously? 

Reviewer #1: Yes

Reviewer #2: I Don't Know

3. Have the authors made all data underlying the findings in their manuscript fully available?

Reviewer #1: No

Reviewer #2: Yes

4. Is the manuscript presented in an intelligible fashion and written in standard English?

Reviewer #1: Yes

Reviewer #2: Yes

5. Review Comments to the Author

Reviewer #1: In this study, the authors used the SEER database to explore the trends in morbidity and mortality of patients with liver metastases from ovarian cancer, and examined the impact of different treatments on overall survival (OS) and cancer-specific survival (CSS). They found a therapeutic benefit of surgery for OC patients with liver metastases. The manuscript has the following problems:

1. The authors demonstrated the effect of surgery on OS and CSS in ovarian cancer patients through their study, but they only presented the HR and 95% CI in the data, and did not provide specific values for OS to clarify the extent of benefit for patients. It is hoped that these data will be added.

2. The authors are requested to carefully review and revise the manuscript for language clarity and accuracy, such as in line 233 where they stated "there was a significant decreasing trend in the OS/CSS between 2010 and 2019. This could be attributed to timely intervention, and advancements in therapeutic approaches". This sentence seems to be contradictory and logically incorrect.

3. In this manuscript, the authors found that surgery can benefit patients, but in the subgroup analysis, different surgical approaches did not seem to affect patient OS and CSS. Please discuss this finding and its implications.

4.Do the liver metastasis cases included in the study refer to initially treated or recurrent patients? An explanation of this would be appreciated.

Reviewer #2: 1.The authors concluded that only receiving surgery and chemotherapy were protective factors for OS and CSS of patients. However, for patients with advanced ovarian cancer, surgery combined with chemotherapy is a necessary treatment, and such conclusions fail to provide new references for current clinical decision-making. It is suggested that the authors attempt to construct and validate nomogram models to predict CSS and OS in patients with ovarian cancer liver metastases, providing more personalized and practical clinical support.

2.The manuscript concluded that the morbidity and mortality rates among patients with liver metastases originating from ovarian cancer showed a downward trend. However, this is a trend analysis for the entire population, and further stratification could be attempted to help readers gain a deeper understanding of the trends in the incidence and mortality of ovarian cancer combined with liver metastases. For example, stratified by age, tumor grade, and treatment modality could provide more nuanced insights.

3.In the "Statistical Analysis" part, from line 130 to line 133, the authors used relevant models to identify potential covariates related to OS and CSS and built a model 2 adjusted for covariates. However, the exact method of constructing model 2 was not elaborated upon. It is recommended that the authors supplement this section with a detailed explanation of the methods used to adjust for covariates.

6. PLOS authors have the option to publish the peer review history of their article (what does this mean?). If published, this will include your full peer review and any attached files.

Reviewer #1: No

Reviewer #2: No

---

## [Author Response · Author response to Decision Letter 0]

30 Nov 2023

Reviewer #1: In this study, the authors used the SEER database to explore the trends in morbidity and mortality of patients with liver metastases from ovarian cancer, and examined the impact of different treatments on overall survival (OS) and cancer-specific survival (CSS). They found a therapeutic benefit of surgery for OC patients with liver metastases. The manuscript has the following problems:

1. The authors demonstrated the effect of surgery on OS and CSS in ovarian cancer patients through their study, but they only presented the HR and 95% CI in the data, and did not provide specific values for OS to clarify the extent of benefit for patients. It is hoped that these data will be added.

Answer: Thanks very much for your advice. According to your suggestion, we have added the Kaplan-Meier curves about the impact of different treatment methods on OS (Supplemental Figure 1) and CSS (Supplemental Figure 2) in this study. The result also indicated that only received surgery had a higher OS (Supplemental Figure 1a) and CSS (Supplemental Figure 2a). Surgical method also appeared to affect patient survival, with pelvic exenteration were found to have a higher OS (Supplemental Figure 1b) and CSS than others (Supplemental Figure 2b). Furthermore, patients who received radiotherapy may have lower OS (Supplemental Figure 1c) and CSS (Supplemental Figure 2c). The impact of chemotherapy on prognosis of patients with ovarian cancer was obvious, leading to significantly improved OS (Supplemental Figure 1d) and CSS (Supplemental Figure 2d) compared to those who do not receive this treatment. Overall, surgery may be benefit for OS and CSS of patients with ovarian cancer.

2. The authors are requested to carefully review and revise the manuscript for language clarity and accuracy, such as in line 233 where they stated "there was a significant decreasing trend in the OS/CSS between 2010 and 2019. This could be attributed to timely intervention, and advancements in therapeutic approaches". This sentence seems to be contradictory and logically incorrect.

Answer: Thanks very much for your suggestion. We are sorry for the wrong description in the manuscript. We have modified the content as follows: Additionally, there was a significant decreasing trend in the all-cause mortality/cancer-specific mortality between 2010 and 2019. This could be attributed to timely intervention, and advancements in therapeutic approaches [16]. Although there was no statistical significance in certain subgroups, all incidence rate, all-cause mortality, cancer specific mortality trend showed a tendency to a decrease in ovarian cancer with liver metastases when stratified by age, tumor grade, and treatment modality. Please see the revised manuscript. 

3. In this manuscript, the authors found that surgery can benefit patients, but in the subgroup analysis, different surgical approaches did not seem to affect patient OS and CSS. Please discuss this finding and its implications.

Answer: Thanks for your suggestion. According to your advice, we have added the content in the Discussion section: Furthermore, in all subgroup analyses, only receiving surgery and chemotherapy also were beneficial for OS/CSS for patients. Similar to the findings in the general population, we observed no significant impact of different surgical method on patients' OS and CSS in subgroups without other distant metastases (Subgroup Ⅰ), with distant metastases to the bone, lung, brain or other organs (Subgroup Ⅱ), and with lung metastasis (Subgroup Ⅳ) (P>0.05). Notably, when comparing to Debulking as a reference, the absence of surgery posed a risk factor for both OS and CSS within these specific subgroups. This observation may be attributed to the limited sample size of our study. It is worth noting that oophorectomy with omentectomy, pelvic exenteration and natural orifice surgery were related to CSS for ovarian cancer patients with liver metastases who combined bone metastasis. These findings may also imply that the type of surgery has an impact on the prognosis of diverse patients. Further prospective studies with larger sample size should be conducted to confirm our findings.

4. Do the liver metastasis cases included in the study refer to initially treated or recurrent patients? An explanation of this would be appreciated.

Answer: Thanks for your suggestion. The cases of liver metastasis in the study referred to initially treated patients. This retrospective cohort study used data from the SEER database. Patients who were diagnosed with primary ovarian cancer based on International Classification of Diseases for Oncology codes (ICD-O-3) and had presence of liver metastases at the time of diagnosis were included. All included liver metastasis cases refer to initially treated patients.

Reviewer #2: 1. The authors concluded that only receiving surgery and chemotherapy were protective factors for OS and CSS of patients. However, for patients with advanced ovarian cancer, surgery combined with chemotherapy is a necessary treatment, and such conclusions fail to provide new references for current clinical decision-making. It is suggested that the authors attempt to construct and validate nomogram models to predict CSS and OS in patients with ovarian cancer liver metastases, providing more personalized and practical clinical support.

Answer: Thanks very much for your advice. We agree with you. According to your comments, we have added the content about the development of online nomograms to predict patients’ OS and CSS. In this study, two online OS (https://dynamic-nomogram-for-predicting-overall-survival--2023.shinyapps.io/DynNomapp/) and CSS (https://dynamic-nomogram-for-predicting-cancer-specific-survival--2023.shinyapps.io/DynNomapp/) predicting nomogram were established, respectively. C-index was used to verify the predicting performance of two nomogram (Table 4). Specifically, C-index of OS nomogram was 0.756 (95% CI: 0.746-0.766), and OS nomogram was 0.752 (95%CI: 0.742-0.763). These findings also indicated that two online nomograms had a good predicting value for OS and CSS in patients with ovarian cancer liver metastases.

2.The manuscript concluded that the morbidity and mortality rates among patients with liver metastases originating from ovarian cancer showed a downward trend. However, this is a trend analysis for the entire population, and further stratification could be attempted to help readers gain a deeper understanding of the trends in the incidence and mortality of ovarian cancer combined with liver metastases. For example, stratified by age, tumor grade, and treatment modality could provide more nuanced insights.

Answer: Thanks very much for your advice. After considering your suggestion, we have the trend analyses stratified by age, tumor grade, and treatment modality. Supplemental Figure 1a-1e also shows the incidence trend of liver metastases in ovarian cancer stratified by age, tumor grade, and treatment modality. Although there was no statistical significance in certain subgroups, all incidence rate trend showed a tendency to a decrease in liver metastases in ovarian cancer stratified by age, tumor grade, and treatment modality. Supplemental Figure 2a-2e also demonstrates a declining trend in the all-cause mortality of liver metastases in ovarian cancer stratified by age, tumor grade, and treatment modality. Similar results were observed for specific mortality of liver metastases in ovarian cancer in different subgroups (Supplemental Figure 3a-3e).

3.In the "Statistical Analysis" part, from line 130 to line 133, the authors used relevant models to identify potential covariates related to OS and CSS and built a model 2 adjusted for covariates. However, the exact method of constructing model 2 was not elaborated upon. It is recommended that the authors supplement this section with a detailed explanation of the methods used to adjust for covariates.

Answer: Thanks for your suggestion. We have modified the content in the "Statistical Analysis" part: Univariate Cox proportional risk model was utilized to identify potential covariates associated with OS (P<0.05), while the univariate competing risk model was employed to identify potential covariates associated with CSS (P<0.05). Subsequently, OS served as the outcome variable and different treatment methods were considered as independent variables, two models were established to investigate the association between different treatment methods and OS in ovarian cancer patients with liver metastases. Model 1 was univariate Cox proportional risk model (unadjusted covariates). Model 2 was multivariate Cox proportional risk model (adjusted all potential covariates associated with OS). Similarly, to assess the impact of various treatment methods on CSS, we developed univariate and multivariate competing risk models with CSS as the outcome and different treatment methods as independent variables. Model 1 was univariate competing risk model (unadjusted covariates). Model 2 was multivariate competing risk model (adjusted all potential covariates associated with CSS). Please see the revised manuscript.

---

## [Editor Report · Decision Letter 1]

2 Jan 2024

PONE-D-23-22973R1Effect of different treatment modalities on ovarian cancer patients with liver metastases: a retrospective cohort study based on SEERPLOS ONE

Dear Dr. Liu,

Thank you for submitting your manuscript to PLOS ONE. After careful consideration, we feel that it has merit but does not fully meet PLOS ONE’s publication criteria as it currently stands. Therefore, we invite you to submit a revised version of the manuscript that addresses the points raised during the review process.

We look forward to receiving your revised manuscript.

Kind regards,

Federico Ferrari, MD, PhD

Academic Editor

PLOS ONE

Journal Requirements:

Additional Editor Comments:

Dear Authors,

I read with interest your study.

As a further comment I would like to understand the treatment modalities you have included in your analysi. Do you have considered only Primary debulking surgery and excluded Neoadjuvant chemotherapy followed by interval debulking surgery? If this is the case you should state about it in the discussion, in fact resection of liver metastasis is feasible also after chemotherapy. Otherwise if you included these patients you have to definetly reorganize the manuscript highlighting the treatment modalities. In both the option, I suggest to refer to https://www.ejgo.net/articles/10.31083/j.ejgo4301015, so you can either state this limit or clarify the treatment modalities.

Many thanks.

Federico Ferrari, MD, PhD

---

## [Author Response · Author response to Decision Letter 1]

10 Jan 2024

Dear Authors,

I read with interest your study.

As a further comment I would like to understand the treatment modalities you have 

included in your analysis. Do you have considered only Primary debulking surgery and 

excluded Neoadjuvant chemotherapy followed by interval debulking surgery? If this is 

the case you should state about it in the discussion, in fact resection of liver metastasis 

is feasible also after chemotherapy. Otherwise, if you included these patients you have 

to definetly reorganize the manuscript highlighting the treatment modalities. In both the 

option, I suggest to refer to https://www.ejgo.net/articles/10.31083/j.ejgo4301015, so 

you can either state this limit or clarify the treatment modalities.

Many thanks.

Federico Ferrari, MD, PhD

Answer: Thanks very much for your advice. We agree with you. Since all the data in 

this retrospective cohort study were derived from the SEER database, we could not 

distinguish the sequence of chemotherapy and surgery in patients; that is, we could not 

distinguish between PDS and IDS. This is also one of the limitations described in the 

discussion section. Further prospective studies should be conducted to confirm our 

findings. At the same time, we also describe in the discussion section the fact that 

resection of liver metastases after chemotherapy is also feasible [1]. Please see the 

revised manuscript.

[1] Forte S, Ferrari F, Valenti G, Capozzi VA, Santana BN, Babin G, Guyon F. Liver surgery for 

advanced ovarian cancer: a systematic review of literature. Eur. J. Gynaecol. Oncol. 2022; 43(1): 

64–72. doi: http://doi.org/10.31083/j.ejgo4301015

---

## [Editor Report · Decision Letter 2]

12 Feb 2024

Effect of different treatment modalities on ovarian cancer patients with liver metastases: a retrospective cohort study based on SEER

PONE-D-23-22973R2

Dear Dr. Liu,

We’re pleased to inform you that your manuscript has been judged scientifically suitable for publication and will be formally accepted for publication once it meets all outstanding technical requirements.

Kind regards,

Federico Ferrari, MD, PhD

Academic Editor

PLOS ONE
---

## [Editor Report · Acceptance letter]

21 Feb 2024

PONE-D-23-22973R2 

PLOS ONE

Dear Dr. Liu, 

I'm pleased to inform you that your manuscript has been deemed suitable for publication in PLOS ONE. Congratulations! Your manuscript is now being handed over to our production team.

Kind regards, 

on behalf of

Dr Federico Ferrari 

Academic Editor

PLOS ONE